# DATASET DISTILLATION IN LATENT SPACE

## ABSTRACT

Dataset distillation (DD) is a newly emerging research area aiming at alleviating the heavy computational load in training models on large datasets, as it tries to distill a large dataset into a small and condensed one so that models trained on the distilled dataset can perform comparably with those trained on the full dataset in downstream tasks. Among the previous works in this area, there are three key problems that hinder the performance and availability of the existing DD methods: high time complexity, high space complexity, and low info-compactness. In this work, we simultaneously attempt to settle these three problems by moving the DD processes from conventionally used pixel space to latent space. Encoded by a pretrained generic autoencoder, latent codes in the latent space are naturally info-compact representations of the original images in much smaller sizes. After transferring three mainstream DD algorithms to latent space, we significantly reduce time and space consumption while achieving similar performance, allowing us to distill high-resolution datasets or target at greater data ratio that previous methods have failed. Besides, within the same storage budget, we can also quantitatively deliver more info-compact latent codes than pixel-level images, which further boosts the performance of our methods.

## 1 INTRODUCTION

Due to the rapidly progressing compute capability of modern devices, people are building unprecedentedly large and data-hungry models. For instance, the state-of-the-art text-to-image generative models, Stable Diffusions (Rombach et al., 2022), were pretrained on LAION-5B (Schuhmann et al., 2022), which contains 5.85 billion text-image pairs. Although large models trained on large datasets have achieved fascinating performance in many applications, they are still renowned for the high demands on training time, computing devices, storage budgets, and electricity consumption.

In recent years, dataset distillation (DD) is a newly emerging research topic which tries to lower the aforementioned demands. Inspired by knowledge distillation (Hinton et al., 2014; Bucila et al., 2006; Ba & Caruana, 2014; Romero et al., 2015), DD aims at distilling a full dataset into a much smaller set, so that in specific tasks the models trained on this distilled dataset are expected to perform comparably to those trained on the full dataset. Some DD methods select a subset from the full dataset according to certain rules (Feldman et al., 2013; Welling, 2009; Sener & Savarese, 2018; Aljundi et al., 2019; Zhou et al., 2023), usually referred to as *coreset selection*. Wang et al. (2018) has proposed a prototype method of optimization-based DD[1] by solving a bi-level optimization problem, which has started a new era in this area. By analyzing this prototype, we conclude the main issues in DD into three problems. **Problem 1:** DD involves solving a computationally intensive bi-level optimization problem in a nested-loop manner, which has high time complexity. **Problem 2:** DD has to store the whole computation graph recording the optimization process of the network before finally back-propagating the gradients all the way through the graph to update the distilled dataset, making its space complexity high too. **Problem 3:** As we expect to remain a small data ratio (size of distilled dataset/full dataset), the distilled dataset should be highly info-compact in order to cover the information of the full dataset as much as possible. However, distilling dataset in the original space (*e.g.* pixel space for image datasets) will inevitably condense high-frequency detailed information into limited storage budget, which is usually unnecessary for downstream tasks.

---

[1]In the rest of this paper, dataset distillation methods exclusively refer to the optimization-based ones.

Methods following the prototype of DD (Wang et al., 2018) have usually focused on one or two of the three problems. For example, the three mainstream DD algorithms DC (Zhao & Bilen, 2021a), DM (Zhao & Bilen, 2023) and MTT (Cazenavette et al., 2022) respectively use gradient matching, feature matching and parameter matching to efficiently approximate the bi-level optimization of the prototype, improving P1–P2 yet omitting P3. Some other works factorize the distilled dataset into more info-compact *components* (Kim et al., 2022; Liu et al., 2022; Cazenavette et al., 2023) thus alleviate P3. Nevertheless, since these methods still operate on pixel space using the DD algorithms above, they even induce extra time and space consumption when transforming the components to images and inversely back-propagating gradients from images to components and fail P1–P2. To the best of our knowledge, no previous work has simultaneously settled all the three problems.

Within the field of image generation, another domain fraught with computational complexity, recent breakthroughs in diffusion models (Ho et al., 2020; Song et al., 2021; Dhariwal & Nichol, 2021) have pushed both the performance and the time & space consumption to a new level. Later on, Rombach et al. (2022) have proposed latent diffusion models transferring the diffusing/denoising procedures from pixel space to latent space with the help of a pretrained autoencoder, which largely accelerates the training process while keeping the performance. Inspired by such design, we propose **Latent Dataset Distillation (LatentDD)** which transfers the three mainstream DD algorithms DC, DM and MTT to fully operate on latent codes encoded by the pretrained generic autoencoder provided in latent diffusion models rather than on images, namely LatentDC, LatentDM and LatentMTT. An overview of LatentDD is shown in Figure 1 (b) and (c). Since the latent codes have much smaller size than pixel-level images, LatentDD takes significantly less time and space (both main memory and GPU memory) to run DD algorithms, alleviating P1–P2. Such acceleration and space reduction is only at the cost of marginal performance degradation, as the pretrained autoencoder can roughly keep the distribution of the original images into the latent codes and thus solving DD in latent space is approximately equivalent to that in pixel space (see Section 3.2). As for P3, the latent codes are naturally info-compact representations of the original images since the autoencoder can reconstruct the images from latent codes losing just the subtlest details. Besides, with a fixed data ratio (storage budget) in DD tasks, we can quantitatively store much more latent codes than pixel-level images, which boosts the performance of LatentDD.

In summary, our work is the first to settle all the three problems in DD at the same time. It is also worth mentioning that its fast and space-saving designs enable LatentDD to distill high-resolution datasets. While most of the recent works have been dealing with toy datasets like CIFAR10/100 (Krizhevsky, 2009) while only a few latest ones trying higher resolution like 64 or 128, we roll out our experiments on high-resolution settings starting from 256, and beyond. These challenging experiments in Section 4 have manifested the superiority of LatentDD over previous works.

## 2 RELATED WORK

Before Wang et al. (2018), coreset selection (Feldman et al., 2013; Welling, 2009; Sener & Savarese, 2018; Toneva et al., 2019; Aljundi et al., 2019) was the primary solution for distilling datasets. Since Wang et al. (2018) proposed the prototype of DD, follow-up works started to focus on optimization-based dataset distillation, aiming at solving the three problems mentioned in Section 1. Besides the brief introductions below, we also recommend some surveys (Sachdeva & McAuley, 2023; Geng et al., 2023; Lei & Tao, 2023; Yu et al., 2023) for more details.

**Gradient Matching** Zhao & Bilen (2021a) proposed Dataset Condensation (DC), the first practically plausible DD algorithm which eliminated the clumsy bi-level optimization of the prototype. DC used single-step gradient matching as a surrogate objective to bridge the parameter gap when trained on real/synthetic datasets. Following DC, DSA (Zhao & Bilen, 2021b) attached Differentiable Siamese Augmentation to DC framework. DCC/DSAC (Lee et al., 2022b) enhanced DC/DSA with contrastive signal, matching gradient in an all-class manner instead of a class-wise one.

**Feature Matching** Zhao & Bilen (2023) proposed Distribution Matching (DM), which extracted features from both real/synthetic images via randomly initialized network and matched their mean values class-wisely. As another mainstream DD algorithm, DM largely accelerated DD process by completely avoiding bi-level optimization. IDM (Zhao et al., 2023) improved DM with image

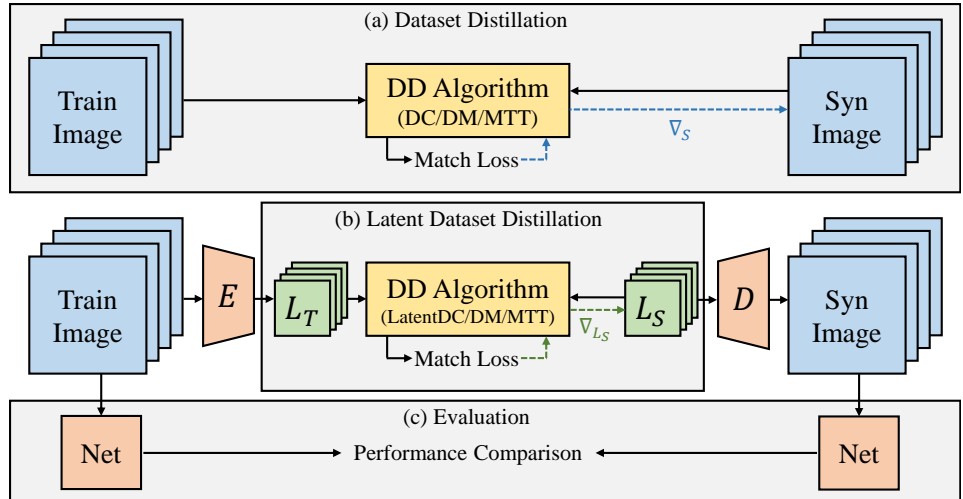

Figure 1: An overview of DD and LatentDD. (a) Procedure of dataset distillation in pixel space, where DD algorithms produce gradients to update synthetic images $\mathcal{S}$. (b) Procedure of dataset distillation in latent space, where DD algorithms directly operate on latent codes and produce gradients to update synthetic latent codes $L_{\mathcal{S}}$. (c) After distillation, networks trained on real training images $\mathcal{T}$ and synthetic images $\mathcal{S}$ will be compared.

partitioning and trained feature extractors. Similar to DM, CAFE (Wang et al., 2022) also compared the mean values of multi-layer features, yet equipped with an extra discrimination loss.

**Parameter Matching** Cazenavette et al. (2022) proposed the third DD algorithm Matching Training Trajectories (MTT). It reduced the accumulated parameter error in gradient matching methods by matching model parameters after long-term training trajectories. After MTT, Li et al. (2023) pruned hard-to-match parameters, FTD (Du et al., 2023) regularized flat trajectories during the buffer phase, and TESLA (Cui et al., 2023) improved MTT by lowering its space complexity.

**Optimization** Besides designing DD algorithms, some other works attempted to enhance these algorithms with optimization techniques, including kernel method (Nguyen et al., 2021a;b; Loo et al., 2022; Zhou et al., 2022), label learning (Bohdal et al., 2020; Cui et al., 2023), model augmentation (Zhang et al., 2023), clustering (Liu et al., 2023) and calibration (Zhu et al., 2023).

**Factorization** Factorization is another research direction orthogonal to designing DD algorithms. It aims at factorizing distilled images into more info-compact components. Specific strategies include image partitioning (Kim et al., 2022; Zhao et al., 2023) and factorizing images into latent codes + decoders (Liu et al., 2022; Deng & Russakovsky, 2022; Lee et al., 2022a; Cazenavette et al., 2023). However, all these methods had to repeatedly restore the pixel-level images before sending them into DD algorithms, and back-propagate the gradients from the images to the components, resulting in heavy time & space overhead. On the contrary, as a factorization method, our LatentDD instead directly operates in latent space, which largely reduces time & space consumption.

## 3 LATENT DATASET DISTILLATION

### 3.1 PROBLEM DEFINITION

Suppose we have a large dataset $\mathcal{T} = \{(x_i, y_i)\}_{i=1}^{|\mathcal{T}|}$ to be distilled, which consists of real pairs of datum $x_i \in \mathbb{R}^d$ and class label $y_i \in \{0, \ldots, C-1\}$ where $d$ is the dimension of the data and $C$ is the number of classes. The goal of dataset distillation is to seek a condensed small dataset $\mathcal{S} = \{(\tilde{x}_i, \tilde{y}_i)\}_{i=1}^{|\mathcal{S}|}$ including synthetic pairs of datum $\tilde{x}_i$ and label $\tilde{y}_i$, and $|\mathcal{S}| \ll |\mathcal{T}|$. Conventionally the synthetic data follow the form of real data (*i.e.* $\tilde{x}_i \in \mathbb{R}^d$) and so are the labels (*i.e.*

$\tilde{y}_i \in \{0, \ldots, C-1\}$). However, there are also some previous works exploring more effective forms of data via factorization, or labels via label learning, as long as the distilled dataset $\mathcal{S}$ does not exceed a predefined storage budget.

With the distilled dataset $\mathcal{S}$, we expect that models trained on it will achieve comparable performance with those trained on the real dataset $\mathcal{T}$ in downstream tasks. Formally,

$$\mathcal{S}^* = \arg\min_{\mathcal{S}} \|\mathcal{L}^{\mathcal{T}}(\theta^{\mathcal{S}}) - \mathcal{L}^{\mathcal{T}}(\theta^{\mathcal{T}})\| \quad \text{subject to} \quad \begin{cases} \theta^{\mathcal{S}} = \arg\min_{\theta} \mathcal{L}^{\mathcal{S}}(\theta) \\ \theta^{\mathcal{T}} = \arg\min_{\theta} \mathcal{L}^{\mathcal{T}}(\theta) \end{cases}, \tag{1}$$

where $\theta^{\mathcal{S}}$ and $\theta^{\mathcal{T}}$ are models trained on $\mathcal{S}$ and $\mathcal{T}$, and $\mathcal{L}^{\mathcal{S}}$, $\mathcal{L}^{\mathcal{T}}$ are respectively a certain objective (loss function) when evaluated on the two datasets. Since models trained on $\mathcal{S}$ are unlikely to outperform those trained on $\mathcal{T}$, we usually seek $\mathcal{S}^*$ that performs the best:

$$\mathcal{S}^* = \arg\min_{\mathcal{S}} \mathcal{L}^{\mathcal{T}}(\theta^{\mathcal{S}}) \quad \text{subject to} \quad \theta^{\mathcal{S}} = \arg\min_{\theta} \mathcal{L}^{\mathcal{S}}(\theta). \tag{2}$$

## 3.2 FROM PIXEL TO LATENT SPACE

The prototype of DD straightforwardly solves Eq. (2) in a bi-level optimization manner, and later three mainstream DD algorithms DC, DM and MTT have been proposed to efficiently solve Eq. (2) by optimizing surrogate objectives. Primarily focusing on image classification, all the previous works have distilled datasets in pixel space. When solving Eq. (2) with DD algorithms, data in both $\mathcal{T}$ and $\mathcal{S}$ are in the form of the original pixel-level images (*i.e.* $x_i, \tilde{x}_i \in \mathbb{R}^{3 \times H \times W}$, as three-channel images with size $(H, W)$). Although some previous works have proposed factorization methods which distill more info-compact components instead of images, they are still based on DD algorithms in pixel space. As pixel-level images usually consist of low-frequency information, which includes the contents that are truly necessary to downstream tasks like image classification, *and* high-frequency information including fine and subtle details and even noises. Taking both parts of information into account when running DD algorithms has brought considerable overhead in time and space consumption, especially when distilling high-resolution datasets or aiming at a higher data ratio. Also, with the useless high-frequency information occupying some of the storage budget, the less info-compact synthetic images fail to reach better scores in classification tasks.

In this work, we attempt to transfer the DD algorithms to directly operate in latent space rather than pixel space. Suppose we have an autoencoder $\{\mathcal{E}(\cdot), \mathcal{D}(\cdot)\}$ pretrained on a large dataset to ensure its generalization ability to encode and reconstruct any image with its encoder $\mathcal{E}(\cdot)$ and decoder $\mathcal{D}(\cdot)$. Specifically, from a real image $x_i \in \mathbb{R}^{3 \times H \times W}$ in $\mathcal{T}$, the encoder can encode it into a latent code $l_i \in \mathbb{R}^{C \times \frac{H}{f} \times \frac{W}{f}}$ as a $C$-channel feature map with a downsampling factor $f$, and the settings of $C$ and $f$ ensure that the size of $l_i$ is smaller than $x_i$ (*i.e.* $C < 3 \cdot f^2$). Then, from a latent code $l_i$, the decoder can decode it into a reconstructed image $x_i' \in \mathbb{R}^{3 \times H \times W}$ back to the original size. If the autoencoder is well trained, the reconstructed image $x_i'$ should be roughly the same as the original image $x_i$, with acceptable minor loss of details and noises. In this way, the latent code $l_i$ can naturally serve as an info-compact representation of the original image $x_i$ since most of the necessary information needed to reconstruct $x_i$ has been encoded into $l_i$. Therefore, if we can use latent codes instead of images in DD algorithms, it will improve DD processes by simultaneously alleviate the three problems mentioned in Section 1.

Rombach et al. (2022) has proposed latent diffusion models equipped with such a pretrained generic autoencoder, which successfully transfer the time-consuming denoising processes to latent space. In our LatentDD, we utilize this off-the-shelf autoencoder as a converter between images in pixel space and latent codes in latent space. By default, this autoencoder is capable of encoding any image with any resolution into a latent code with $C = 4$ channels and a downsampling factor $f = 8$. For instance, an RGB image of resolution $512 \times 512$ will be encoded into a latent code of size $4 \times 64 \times 64$. By these settings, a latent code is only $1/48$ of the original image w.r.t. the number of parameters, which is highly info-compact. Nevertheless, before transferring DD algorithms to latent space, we still need to verify that this autoencoder will not break the original distribution of the images, as keeping this distribution is critical to image classification tasks. In a preliminary experiment, we randomly select 20 images from *Bird*, a subset of ImageNet (see Section 4.1 for details), encode them into latent codes, and respectively show the Euclidean distance matrices of

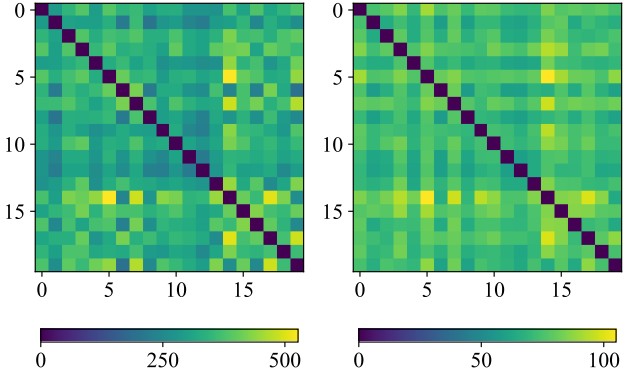
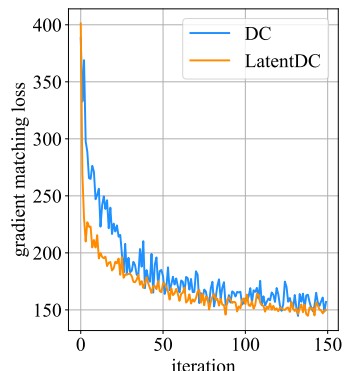

Figure 2: The Euclidean distance matrices of 20 randomly sampled images (left) from dataset *Bird* and their corresponding latent codes (right).

Figure 3: MSE gradient matching loss of DC and LatentDC in the first 150 iterations.

these images and latent codes in Figure 2. From the two heatmaps, we may conclude that the latent codes approximately remain the distribution of the original images, as the relative distances among these latent codes strongly correlate with those among the images. As a result, the learned classification hyperplanes in latent space can correspond to the hyperplanes in pixel space, making DD in latent space and pixel space approximately equivalent. Such equivalence is also empirically validated by ablation study in Section 4.3, where distilling the same number of images/latent codes renders similar performance. Based on Eq. (2), DD in latent space can be formally written as

$$L_{\mathcal{S}}^* = \arg\min_{L_{\mathcal{S}}} \mathcal{L}^{L_{\mathcal{T}}}(\theta^{L_{\mathcal{S}}}) \quad \text{subject to} \quad \theta^{L_{\mathcal{S}}} = \arg\min_{\theta} \mathcal{L}^{L_{\mathcal{S}}}(\theta), \tag{3}$$

where $L_{\mathcal{T}} = \{(\mathcal{E}(x_i), y_i)\}_{i=1}^{|\mathcal{T}|}$ and $L_{\mathcal{S}} = \{(\tilde{l}_i, \tilde{y}_i)\}_{i=1}^{|\mathcal{S}|}$ are respectively the set of real and synthetic latent codes, and we remain $\tilde{y}_i \in \{0, \ldots, C-1\}$ as class labels. Before running LatentDD algorithms, $L_{\mathcal{T}}$ is precomputed and stored into the main memory for fast retrieval, and $L_{\mathcal{S}}$ is initialized with randomly selected real latent codes. After obtaining $L_{\mathcal{S}}^*$ via LatentDD, we can reconstruct the synthetic image dataset as $\mathcal{S}^* = \{(\mathcal{D}(\tilde{l}_i), \tilde{y}_i)\}_{i=1}^{|\mathcal{S}|}$ for downstream tasks, as depicted in Figure 1.

As some previous works (Liu et al., 2022; Kim et al., 2022; Deng & Russakovsky, 2022) focusing on factorization have proved, the fixed data ratio will severely confine the performance on downstream image classification tasks if we stick to delivering distilled datasets as pixel-level images within a limited storage budget. These works instead deliver info-compact components such as low-resolution 'thumbnails' or a combination of latent codes and decoders within the storage budget, which can be resized or decoded into more images than the same budget can directly store. Our LatentDD has followed this idea of storing info-compact components, as we deliver $n \cdot 3f^2/C$ latent codes rather than $n$ pixel-level images. It is also worth mentioning that, previous works of latent codes + decoders also train their decoders along with the latent codes during DD processes, thus their decoders can only be exclusively used on specific datasets (or even specific classes of these datasets) and specific resolutions. So they have to store the decoders as a part of the distilled datasets to be delivered. On the contrary, since our pretrained autoencoder is generic for any image and any resolution, and is also publicly available online, it only takes $O(1)$ space to store the decoder when we distill $N$ datasets instead of $O(N)$ as in previous works. Hence our decoder will averagely take negligible space if we distill many datasets and may be excluded from the storage budget like an unparameterized resizing operation that can be applied to any image.

### 3.3 LATENT DATASET DISTILLATION ALGORITHMS

Dataset Condensation (DC), Distribution Matching (DM) and Matching Training Trajectories (MTT) are three mainstream DD algorithms, which solve the DD problem in Eq. (2) with surrogate objectives of gradient matching, feature matching and parameter matching respectively. We show how to seamlessly transfer these algorithms to latent space as below.

### 3.3.1 LATENTDC

DC (Zhao & Bilen, 2021a) is designed based on the observation that, if the model $\theta^{\mathcal{S}}$ trained on distilled dataset $\mathcal{S}$ has similar parameters with $\theta^{\mathcal{T}}$ trained on real dataset $\mathcal{T}$, it will be certain to perform comparably when evaluated on test set. Such resembling parameters can be achieved by matching the gradients of network parameters $\nabla_\theta$ induced by training the same model $\theta_t$ (at timestep $t$) on $\mathcal{S}$ and $\mathcal{T}$. Formally,

$$\mathcal{S}^* = \arg\min_{\mathcal{S}} \mathbb{E}_{\theta_0 \sim P_{\theta_0}} \big[ \sum_{t=0}^{T-1} D(\nabla_\theta \mathcal{L}^{\mathcal{S}}(\theta_t), \nabla_\theta \mathcal{L}^{\mathcal{T}}(\theta_t)) \big] \quad \text{subject to} \quad \theta_{t+1} \leftarrow \theta_t - \eta \nabla_\theta \mathcal{L}^{\mathcal{S}}(\theta_t), \quad (4)$$

where DC randomly initializes network parameters $\theta_0$ and repeatedly matches the two groups of gradients with a cosine-like gradient matching loss $D(\cdot, \cdot)$ along the $T$-step training process of $\theta_t$ on synthetic dataset $\mathcal{S}$ and classification criterion $\mathcal{L}$ (usually a cross-entropy loss).

To run DC in latent space instead of pixel space, the most essential modification is that we have to replace the network $\theta$ operating on pixel-level images with a new network $\tilde{\theta}$ operating on latent codes. Since the latent codes $l \in \mathbb{R}^{C \times \frac{H}{f} \times \frac{W}{f}}$ still remain a 2D spatial structure just as images, commonly used convolutional architectures are also applicable to them if only we accordingly change the channel numbers and feature map sizes. Besides, as the size of $l$ is much smaller than images, we can use shallower networks with fewer layers as $\tilde{\theta}$. For instance, while distilling image datasets of resolution 256 conventionally uses ConvNetD6 with depth 6, we apply ConvNetD3 as $\tilde{\theta}$ when $f = 8$ and ConvNetD4 when $f = 4$, just as dealing with images of the corresponding resolution (refer to Section 4.1 for details). Being able to reduce the size of networks used in DD is one of the main reasons that our latent version of DD algorithms takes much less time and space to run. With $\tilde{\theta}$, our LatentDC can be formulated as

$$L_{\mathcal{S}}^* = \arg\min_{L_{\mathcal{S}}} \mathbb{E}_{\tilde{\theta}_0 \sim P_{\tilde{\theta}_0}} \big[ \sum_{t=0}^{T-1} D(\nabla_{\tilde{\theta}} \mathcal{L}^{L_{\mathcal{S}}}(\tilde{\theta}_t), \nabla_{\tilde{\theta}} \mathcal{L}^{L_{\mathcal{T}}}(\tilde{\theta}_t)) \big] \quad \text{subject to} \quad \tilde{\theta}_{t+1} \leftarrow \tilde{\theta}_t - \eta \nabla_{\tilde{\theta}} \mathcal{L}^{L_{\mathcal{T}}}(\tilde{\theta}_t), \quad (5)$$

where we follow Kim et al. (2022) to (1) update $\tilde{\theta}$ using real latent codes $L_{\mathcal{T}}$ instead of synthetic $L_{\mathcal{S}}$ as the gradients will quickly vanish if trained on the latter and (2) use an MSE-like gradient matching loss for $D(\cdot, \cdot)$ which better fits training on real datasets. In Figure 3, we illustrate the gradient matching losses of both DC and LatentDC in the first 150 iterations when distilling *Bird* into one image/latent code per class (see Section 4.3), where the loss of LatentDC decreases slightly faster, and more steadily than its counterpart in pixel space.

Based on DC, Zhao & Bilen (2021b) add Differentiable Siamese Augmentation (DSA) during both training and evaluation stages, which has become a standard technique in following DC-based works. However, according to our preliminary experiments, only two transformations (crop, cutout) used in DSA can also be applied to latent codes, while the others (color, scale, rotate, flip) on latent codes will unexpectedly affect the quality of the decoded images. Hence, during the training stage in Eq. (5) we do not apply DSA, but still augmenting the pixel-level images decoded from latent codes in the evaluation stage (see Appendix A.2). Such strategy is also adopted by LatentDM and LatentMTT. We suppose that designing a set of feasible transformations for augmenting latent codes is another topic worth researching on, yet we leave this part for future work.

### 3.3.2 LATENTDM

Unlike the other two algorithms DC and MTT, DM (Zhao & Bilen, 2023) is designed aiming at totally eliminate the computationally intensive bi-level optimization. It updates the distilled dataset $\mathcal{S}$ so that the empirical estimate of maximum mean discrepancy (MMD) is minimized between each class of $\mathcal{S}$ and $\mathcal{T}$:

$$\mathcal{S}^* = \arg\min_{\mathcal{S}} \mathbb{E}_{\theta \sim P_\theta} \big[ \sum_{c \in \{0,\dots,C-1\}} \| \frac{1}{|\mathcal{T}_c|} \sum_{x_i \in \mathcal{T}_c} \phi_\theta(x_i) - \frac{1}{|\mathcal{S}_c|} \sum_{\tilde{x}_i \in \mathcal{S}_c} \phi_\theta(\tilde{x}_i) \|^2 \big], \quad (6)$$

where $\mathcal{T}_c, \mathcal{S}_c$ are the subsets of class $c$, and $\phi_\theta$ is an embedding function based on randomly initialized networks $\theta$. Similar to LatentDC, we can also transfer DM to LatentDM by using embedding networks $\tilde{\theta}$ of proper architecture:

$$L_{\mathcal{S}}^* = \arg\min_{L_{\mathcal{S}}} \mathbb{E}_{\tilde{\theta} \sim P_{\tilde{\theta}}} \big[ \sum_{c \in \{0,\dots,C-1\}} \| \frac{1}{|L_{\mathcal{T}_c}|} \sum_{l_i \in L_{\mathcal{T}_c}} \phi_{\tilde{\theta}}(l_i) - \frac{1}{|L_{\mathcal{S}_c}|} \sum_{\tilde{l}_i \in L_{\mathcal{S}_c}} \phi_{\tilde{\theta}}(\tilde{l}_i) \|^2 \big], \quad (7)$$

where the embedding function $\phi_{\tilde{\theta}}$ now extracts embeddings from latent codes rather than images.

### 3.3.3 LATENTMTT

MTT (Cazenavette et al., 2022) is designed to alleviate the issue of the accumulated parameter error caused by the difference between gradients $\nabla_\theta \mathcal{L}^\mathcal{S}(\theta_t)$ and $\nabla_\theta \mathcal{L}^\mathcal{T}(\theta_t)$ in Eq. (4) of DC. Starting from the same network $\theta_t^\mathcal{T}$ that has been trained on the real dataset $\mathcal{T}$ for $t$ steps, it matches the parameters of two networks $\theta_{t+N}^\mathcal{S}$ and $\theta_{t+M}^\mathcal{T}$ respectively after a student trajectory which trains on $\mathcal{S}$ for $N$ steps and an expert trajectory which trains on $\mathcal{T}$ for $M$ steps:

$$\mathcal{S}^* = \arg\min_{\mathcal{S}} \mathbb{E}_{t \in \{0,...,T^+\}} \left[ \frac{\|\theta_{t+N}^\mathcal{S} - \theta_{t+M}^\mathcal{T}\|_2^2}{\|\theta_t^\mathcal{T} - \theta_{t+M}^\mathcal{T}\|_2^2} \right], \tag{8}$$

where the starting step $t$ is sampled within the limit of a maximum starting step $T^+$ since the later part of the a training trajectory is less informative. By matching parameters through multi-step trajectories instead of single-step gradients, MTT generally outperforms DC at the expense of greater time and space consumption. Just as in LatentDC and LatentDM, we can modify MTT into LatentMTT by moving both the expert and the student trajectories into the latent space:

$$L_\mathcal{S}^* = \arg\min_{L_\mathcal{S}} \mathbb{E}_{t \in \{0,...,T^+\}} \left[ \frac{\|\tilde{\theta}_{t+N}^{L_\mathcal{S}} - \tilde{\theta}_{t+M}^{L_\mathcal{T}}\|_2^2}{\|\tilde{\theta}_t^{L_\mathcal{T}} - \tilde{\theta}_{t+M}^{L_\mathcal{T}}\|_2^2} \right], \tag{9}$$

where we also pre-buffer the expert trajectories in latent space for faster run-time matching, similar to MTT in pixel space.

## 4 EXPERIMENTS

### 4.1 EXPERIMENTAL SETTINGS

**Datasets** To fully illustrate the capability of our LatentDD methods, we conduct experiments on high-resolution datasets starting from 256, which is higher than the maximum resolution of almost all the previous works. Specifically, we take five subsets of ImageNet (Deng et al., 2009), namely *Bird* (ImageSquawk), *Fruit* (ImageFruit), *Woof* (ImageWoof), *Cat* (ImageMeow) and *Nette* (ImageNette), where Woof and Nette are online resources[2] and the other three comes from Cazenavette et al. (2022). Our experiments cover different settings of DD aiming at image classification as the downstream task, including resolution 256 or 512, and image per class (IPC) 1 or 10.

**Baselines** We include previous state-of-the-art methods based on each of the three mainstream DD algorithms DC, DM and MTT. For DC, we include DSA (Zhao & Bilen, 2021b) with augmentation, IDC (Kim et al., 2022) which partitions images and GLaD DC (Cazenavette et al., 2023) based on GAN prior. For DM, we include the original DM (Zhao & Bilen, 2023) and GLaD DM Cazenavette et al. (2023). Finally for MTT, we include MTT (Cazenavette et al., 2022), GLaD MTT (Cazenavette et al., 2023) and FTD (Du et al., 2023) with flattened expert trajectories.

**Choice of Autoencoder** Although we may use any autoencoder that remains the distribution of the original images among the latent codes, we practically choose the one as a part of Stable Diffusion v1.4[3] as the pretrained generic autoencoder. This autoencoder is capable of encoding any image with a downsampling factor $f = 8$. Though this autoencoder can encode images of any resolution, it was originally trained on resolution 512 and we have observed that more details will be lost in the reconstructed images if used on lower resolution (refer to Appendix B.1). Therefore, we also propose a preprocessing procedure, in which we first upsample the original pixel-level dataset $\mathcal{T}$ by a factor of two, and then encode the resized dataset into latent codes $L_\mathcal{T}$, resulting in a downsampling factor $f = 4$. Before evaluation, we also follow a postprocessing procedure where we downsample the decoded images by a factor of two, restoring the synthetic dataset $\mathcal{S}$ at the original resolution. By default, we apply these procedures of $f = 4$ in the experiments below unless otherwise specified.

For other implementation details including hyperparameter settings and the training/evaluation protocols, please refer to Appendix A.

---

[2] https://github.com/fastai/imagenette
[3] https://huggingface.co/CompVis/stable-diffusion-v-1-4-original

Table 1: Quantitative results of dataset distillation experiments on ImageNet subsets. LatentDD algorithms follow the setting of $f = 4$, where LPC $= 12 \times$ IPC. The mean classification accuracy among five times of evaluations is reported. The results marked as a hyphen - indicate that the methods have run out of 24GB GPU memory during the experiments.

| Algo. | Method | Res. 256 | | | | | | | Res. 512 | |
| | | IPC 1 | | | | | IPC 10 | | IPC 1 | |
| | | Bird | Fruit | Woof | Cat | Nette | Bird | Fruit | Bird | Fruit |
|---|---|---|---|---|---|---|---|---|---|---|
| DC | DSA | 30.52 | 20.28 | 22.12 | 22.20 | 34.40 | 45.52 | 30.48 | 25.44 | 16.40 |
| | IDC | 36.28 | 24.60 | 25.64 | 27.68 | 48.16 | 64.28 | 39.68 | 28.68 | 20.12 |
| | GLaD DC | 30.32 | 19.56 | 21.84 | 22.24 | 34.44 | 46.04 | 32.60 | 25.80 | 16.72 |
| | **LatentDC** | **46.72** | **30.12** | **28.96** | **38.08** | **55.92** | **80.44** | **51.60** | **47.52** | **29.68** |
| DM | DM | 27.64 | 19.48 | 20.04 | 21.16 | 32.08 | 41.56 | 28.12 | 28.28 | 19.72 |
| | GLaD DM | 28.84 | 21.28 | 21.28 | 20.52 | 32.40 | - | - | 29.32 | 20.68 |
| | **LatentDM** | **47.08** | **30.68** | **28.00** | **36.28** | **56.08** | **77.20** | **47.76** | **46.20** | **30.60** |
| MTT | MTT | 35.80 | 21.08 | 24.92 | 24.16 | 40.52 | - | - | - | - |
| | GLaD MTT | 32.20 | 20.48 | 23.00 | 21.44 | 31.84 | - | - | - | - |
| | FTD | 35.96 | 21.76 | 26.32 | 26.60 | 40.96 | - | - | - | - |
| | **LatentMTT** | **52.86** | **37.82** | **39.84** | **41.42** | **62.86** | **78.44** | **52.46** | **52.44** | **36.20** |

Table 2: Time and space (main memory) consumption during the dataset building, buffering and training processes of LatentDD and the baselines, evaluated on a single NVIDIA RTX 4090. A hyphen - indicates that the method has run out of 24GB GPU memory during the experiment.

| Algo. | Method | Bird 256 | | Bird 512 |
| | | IPC 1 | IPC 10 | IPC 1 |
|---|---|---|---|---|
| | Build Dataset | **2min** / 26.1GB | | **2min** / 94.6GB |
| | **Build Latent Dataset** | 10min / **5.0GB** | | 52min / **9.9GB** |
| DC | DSA | 6h 34min | 30h 27min | 25h 17min |
| | IDC | 32h 36min | 40h 26min | 46h 50min |
| | GLaD DC | 15h 15min | 119h 20min | 30h 51min |
| | **LatentDC** | **1h 59min** | **3h 18min** | **3h 21min** |
| DM | DM | 10min | 14min | 51min |
| | GLaD DM | 55min | - | 2h 15min |
| | **LatentDM** | **1min** | **2min** | **1min** |
| MTT | Buffer | | 16h 20min | 48h 50min |
| | FTD Buffer | | 24h 00min | 92h 30min |
| | **Latent Buffer** | | **50min** | **3h 10min** |
| | MTT | 7h 03min | - | - |
| | GLaD MTT | 11h 48min | - | - |
| | FTD | 7h 17min | - | - |
| | **LatentMTT** | **2h 14min** | **4h 16min** | **2h 38min** |

## 4.2 LATENTDD VS. BASELINES

The quantitative results of the comparisons between LatentDD algorithms and baseline methods are shown in Table 1, where the distilled datasets are evaluated on ConvNets whose depths correspond to image resolutions (see Appendix A). In these experiments, our LatentDD algorithms follow the setting of $f = 4$, hence we deliver 12 latents per class (LPC) within the same storage of 1 IPC, and accordingly 120 LPC within 10 IPC. As our LatentDD methods deliver highly info-compact latent codes, they are able to significantly outperform previous works which deliver less info-compact low-resolution thumbnails (IDC) or full-size images (other baselines). In this way, our LatentDD has successfully settled the info-compactness problem (P3) mentioned in Section 1. Additionally, we will show cross-architecture results and depict some distilled samples with three LatentDD algorithms in Appendix B.

Besides, we also list the running time and main memory consumption of the above experiments in Table 2. Tough LatentDD methods spend more time on building the datasets into main memory

Table 3: Quantitative results of dataset distillation where both IPC and LPC are set to 1 or 10.

| Method | Bird 256 | |
| --- | --- | --- |
| | IPC/LPC 1 | IPC/LPC 10 |
| DSA | 30.52 | 45.52 |
| LatentDC | 29.68 | 45.56 |
| DM | 27.64 | 41.56 |
| LatentDM | 26.64 | 45.00 |
| MTT | 35.80 | - |
| LatentMTT | 33.60 | 46.88 |

Table 4: Quantitative results of dataset distillation where LatentDD methods are compared between $f = 4$ and $f = 8$.

| Method | $f$ | Res. 256 IPC 1 | | Res. 512 IPC 1 | |
| --- | --- | --- | --- | --- | --- |
| | | Bird | Fruit | Bird | Fruit |
| LatentDC | 4 | 46.72 | 30.12 | 47.52 | 29.68 |
| | 8 | 67.28 | 38.08 | 69.46 | 37.22 |
| LatentDM | 4 | 47.08 | 30.68 | 46.20 | 30.60 |
| | 8 | 67.40 | 37.60 | 68.22 | 37.20 |
| LatentMTT | 4 | 52.86 | 37.82 | 52.44 | 36.20 |
| | 8 | 66.42 | 45.28 | 69.20 | 42.84 |

since they pre-encode the real datasets into latent codes, the buffering processes of LatentMTT and the training processes of all latent methods have been largely accelerated due to the smaller sizes of both the latent codes and the networks. The time spent on decoding latent codes into images is negligible since it only takes a few seconds once before evaluation. As for the space consumption, it is much more efficient to store latent codes into main memory than images when building datasets, and the low run-time GPU memory costs also allow LatentDD methods to run on higher resolution or greater data ratio. In conclusion, our LatentDD methods have settled the problems of high time & space complexity (P1–P2) in Section 1 as well.

### 4.3 ABLATION STUDIES

We transfer the distillation processes from pixel space to latent space based on the fact that DD in latent space is approximately equivalent to that in pixel space. Along with the theoretical analysis in Section 3.2, we also empirically verify such equivalence by comparing LatentDD methods under the same IPC/LPC rather than the same storage budget. As the results shown in Table 3, LatentDD still achieves comparable performance with the DD algorithms in pixel space. These results also match Figure 3 where LatentDC can converge to a similar level of gradient matching loss to DC.

In Table 4, we additionally provide quantitative results of LatentDD methods with downsampling factor $f = 8$, where LPC $= 48 \times$ IPC. As the number of latent codes has further increased within the same storage budget, LatentDD methods reach even higher scores. However, as we have mentioned in Section 4.1 that the autoencoder is originally pretrained on resolution 512, encoding and reconstructing images with overly low resolution and a large downsampling factor $f$ may induce too much loss of information that they may alter the distribution of the original images and finally counterweight the benefits brought by the quantity of latent codes (see examples and further analysis in Appendix B.1). For instance, while reaching a mean accuracy of 44.38 when running LatentDC with $f = 4$ on *Bird* at low resolution 64 and IPC 1, which is a comparable result at resolution 256 (46.72) or 512 (47.52) in Table 4, we have observed a remarkable drop from 67.28/69.46 to 48.68 with $f = 8$. In practical applications, it is essential to emphasize the need for maintaining this balance according to the characteristics of specific datasets.

### 5 CONCLUSIONS

In this work, we propose a new idea of transferring the dataset distillation (DD) processes from conventionally used pixel space to a more efficient latent space. Such transfer aims at three main problems commonly seen in the area of DD: **P1:** high time complexity; **P2:** high space complexity; and **P3:** low info-compactness, which have not been simultaneously settled in previous works. Practically, we propose LatentDD methods based on the three mainstream DD algorithms in pixel space by utilizing the latent codes encoded with a pretrained generic autoencoder as natural info-compact representations of pixel-level images. Experimental results have validated that LatentDD methods can not only largely reduce time and space consumption and enable these methods on higher resolution or data ratio, but also significantly boost the performance in DD tasks by delivering quantitatively more info-compact latent codes than pixel-level images within the same storage budget.

## REPRODUCIBILITY STATEMENT

In order to reproduce the results reported in this work, please follow the experimental settings introduced in Section 4.1 in the main paper and Appendix A in the appendices. Besides, the source codes of our LatentDD methods have been attached as supplementary material as well.

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

# APPENDIX

## A    IMPLEMENTATION DETAIL

When running our LatentDD algorithms, we generally follow the default settings of the previous works. Besides, we fix some settings across the baselines and our methods to ensure that the comparisons are as fair as possible. In this section of appendices, we will introduce basic settings to reproduce the experimental results.

### A.1    HYPERPARAMETERS

The hyperparameters under different experimental settings are listed in Table 5 below, except for other hyperparameters that have been introduced in our main paper or by previous works. *Base learning rate* can be seen as the learning rate *per latent code*, as the cross-entropy loss $\mathcal{L}$ will be averaged among the latent codes. Hence the real learning rate updating the latent codes is set to

$$\text{learning rate} = \text{base learning rate} \times \text{LPC}. \tag{10}$$

Note that these settings are just used to produce the results reported in this work, they are not guaranteed to be the optimized settings to render the best performance.

Table 5: Hyperparameter settings of the experiments on LatentDD algorithms. A hyphen - indicates that the hyperparameter is not applicable to this algorithm.

| Res. | IPC | $f$ | Hyperparameter | LatentDC | LatentDM | LatentMTT |
|---|---|---|---|---|---|---|
| | | | iteration | 1000 | 1000 | 5000 |
| | | | max starting epoch | - | - | 5 |
| 256 | 1 | 4 | outer loop / expert epoch $M$ | 10 | - | 1 |
| | | | inner loop / student step $N$ | 50 | - | 40 |
| | | | base learning rate | 0.05 | 0.5 | 50 |
| | | | batch size | 64 | 64 | 64 |
| | | | ConvNet depth (train / eval) | | 4 / 6 | |
| | | 8 | outer loop / expert epoch $M$ | 10 | - | 1 |
| | | | inner loop / student step $N$ | 50 | - | 60 |
| | | | base learning rate | 0.05 | 0.1 | 1 |
| | | | batch size | 64 | 64 | 64 |
| | | | ConvNet depth (train / eval) | | 3 / 6 | |
| | 10 | 4 | outer loop / expert epoch $M$ | 10 | - | 1 |
| | | | inner loop / student step $N$ | 50 | - | 80 |
| | | | base learning rate | 0.05 | 0.5 | 10 |
| | | | batch size | 64 | 64 | 64 |
| | | | ConvNet depth (train / eval) | | 4 / 6 | |
| 512 | 1 | 4 | outer loop / expert epoch $M$ | 10 | - | 1 |
| | | | inner loop / student step $N$ | 50 | - | 40 |
| | | | base learning rate | 0.25 | 0.5 | 500 |
| | | | batch size | 32 | 32 | 32 |
| | | | ConvNet depth (train / eval) | | 5 / 7 | |
| | | 8 | outer loop / expert epoch $M$ | 10 | - | 1 |
| | | | inner loop / student step $N$ | 50 | - | 60 |
| | | | base learning rate | 0.05 | 0.1 | 20 |
| | | | batch size | 16 | 16 | 16 |
| | | | ConvNet depth (train / eval) | | 4 / 7 | |

### A.2    TRAINING/EVALUATION PROTOCOL

Following previous works, we adopt ConvNet (Gidaris & Komodakis, 2018) as the network architecture primarily used in the experiments. We set the depth of the ConvNet according to the resolution

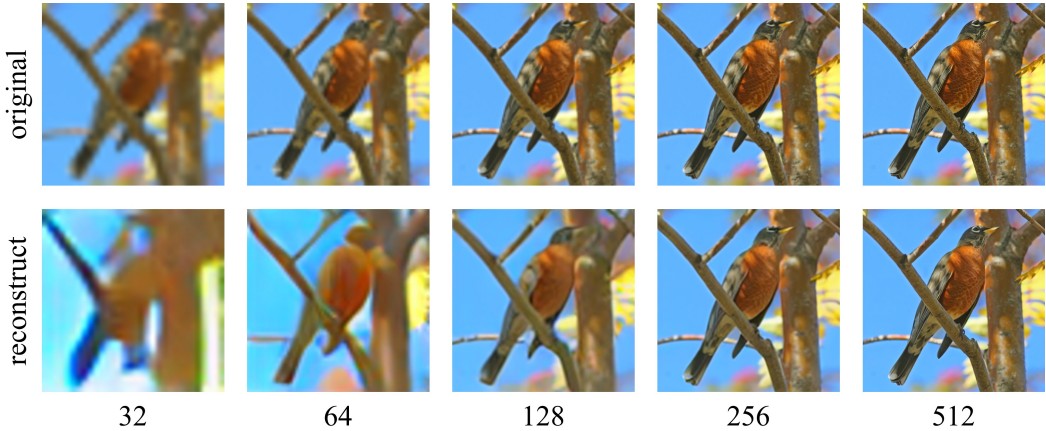

Figure 4: The reconstructed images (encoded from the original images and decoded back by the autoencoder) at different resolutions from 32 to 512.

Table 6: Quantitative results of dataset distillation experiments on ImageNet subsets, evaluated on cross-architecture networks (ResNet18, VGG11 and AlexNet). LatentDD algorithms follow the setting of $f = 4$, where LPC $= 12 \times$ IPC. The mean classification accuracy among five times of evaluations is reported.

| Algo. | Method | Bird 256 IPC 1 | | | |
|---|---|---|---|---|---|
| | | ConvNet | ResNet18 | VGG11 | AlexNet |
| DC | DSA | 30.52 | 14.84 | 20.68 | 24.24 |
| | IDC | 36.28 | 40.24 | 30.20 | 32.24 |
| | GLaD DC | 30.32 | 23.20 | 22.08 | 19.44 |
| | **LatentDC** | **46.72** | **56.00** | **49.32** | **37.56** |
| DM | DM | 27.64 | 13.44 | 17.40 | 23.64 |
| | GLaD DM | 28.84 | 26.24 | 18.24 | 19.08 |
| | **LatentDM** | **47.08** | **56.00** | **47.56** | **37.12** |
| MTT | MTT | 35.80 | 18.72 | 22.36 | 20.84 |
| | GLaD MTT | 32.20 | 39.56 | 23.28 | 19.80 |
| | FTD | 35.96 | 19.28 | 21.96 | 23.92 |
| | **LatentMTT** | **52.86** | **57.76** | **52.96** | **39.64** |

of the latent codes (during training stage) or pixel-level images (during evaluation stage), refer to Table 5 for details. The learning rate updating the network parameters in both stages is set to 0.01.

As explained in Section 3.3.1, we do not apply Differentiable Siamese Augmentation (DSA) during the training processes because many augmenting transformations designed for pixel-level images do not fit for latent codes. However, we use DSA in the evaluation processes in pixel space. Following Kim et al. (2022), we additionally replace cutout with CutMix (Yun et al., 2019), which is a calibration technique to alleviate the over-confident issue of the models trained on limited data.

## B ADDITIONAL RESULTS

In this section of appendices, we illustrate some additional experimental results that have not been shown in the main paper due to the limitation on paper length. These results include preliminary analysis of the autodencoder in Appendix B.1, the performance of LatentDD methods when evaluated on cross-architecture networks in Appendix B.2, and some image samples decoded from the distilled latent codes in Appendix B.3.

## B.1   ANALYSIS OF AUTOENCODER

In Figure 4 we show some examples of reconstructing original images at different resolutions with the autoencoder used in LatentDD. Although the autoencoder can be applied to a variety of resolutions, we observe that the lower the resolution is, the more details will be lost in the reconstruction. Therefore, the experiments in our paper are mainly rolled out with $f = 4$ where we adopt the *upsample-encode-LatentDD-decode-downsample* procedure for a generalized applicability on various resolutions, though in high-resolution scenarios $f = 8$ may produce better results due to the greater quantity of latent codes.

## B.2   CROSS-ARCHITECTURE PERFORMANCE

One of the goals of dataset distillation is that we expect the distilled datasets can achieve good performance on any network architecture, so that these distilled datasets can be utilized on downstream tasks such as neural architecture search (Such et al., 2020). In Table 6, we demonstrate the uniformly good performance of LatentDD algorithms on some cross-architecture networks. As all the baselines are trained on the same architecture used for evaluation (*e.g.* ConvNetD6 for resolution 256), they are more or less overfitted to the specific architecture as a trend of performance drop can be observed when evaluated on other architectures. On the contrary, our LatentDD methods are relatively robust to architecture changes. It is also worth mentioning that the results of LatentDD methods on ConvNet in Table 6 (and also Tables 1, 3 and 4 in the main paper) are already cross-architecture to some extent, since the depths of the ConvNets used for training and evaluation are different.

## B.3   QUALITATIVE RESULTS

For a more comprehensive illustration of the experimental results, we depict some image samples decoded from the distilled latent codes on the ImageNet subset *Bird* (with classes *peacock*, *flamingo*, *macaw*, *pelican*, *king penguin*, *bald eagle*, *toucan*, *ostrich*, *black swan*, *cockatoo*) by LatentDC, LatentDM and LatentMTT in Figures 5 and 6.

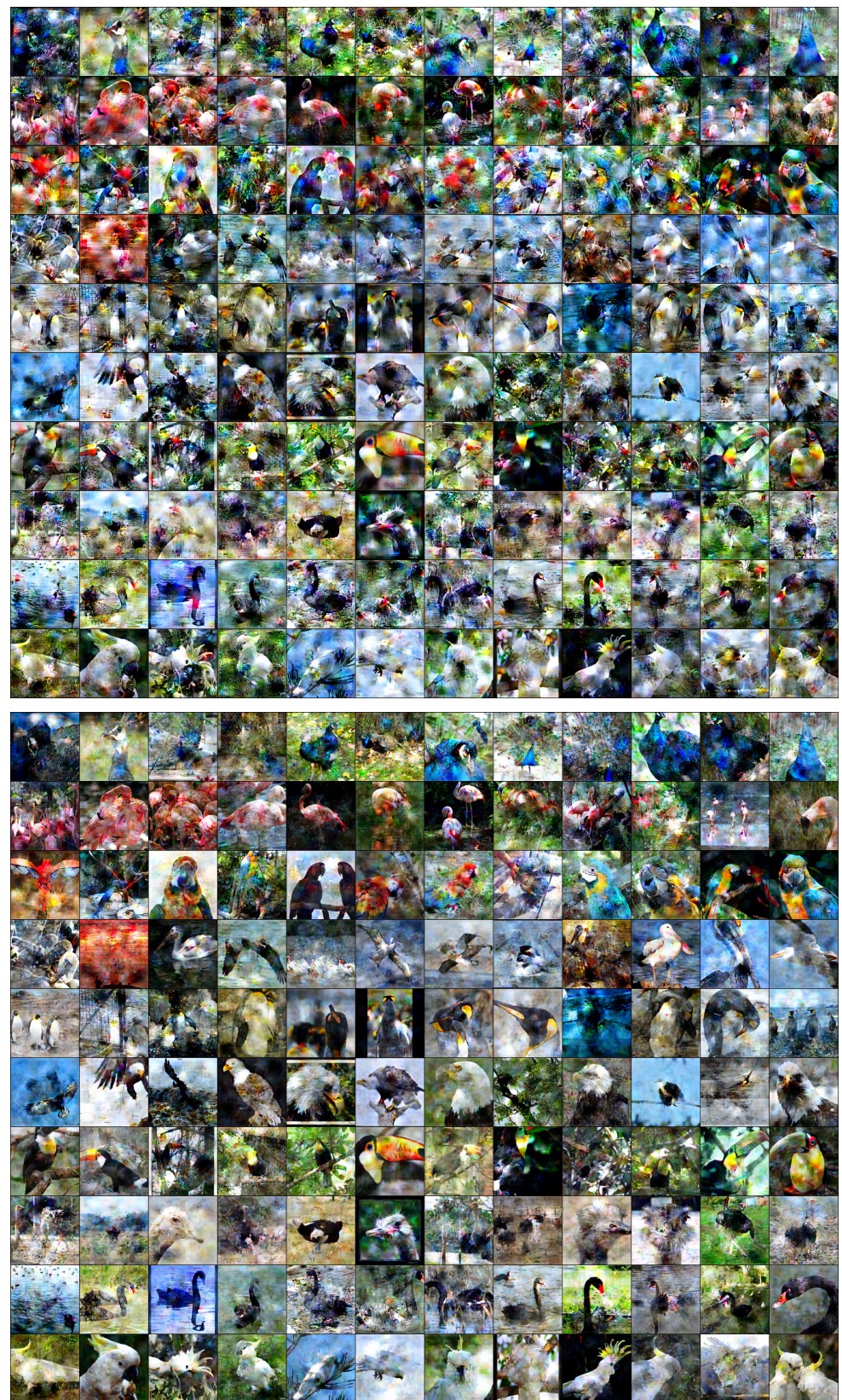

Figure 5: The images decoded from the distilled latent codes on *Bird*, resolution 256, IPC 1, by LatentDC (up) and LatentDM (bottom) with $f = 4$.

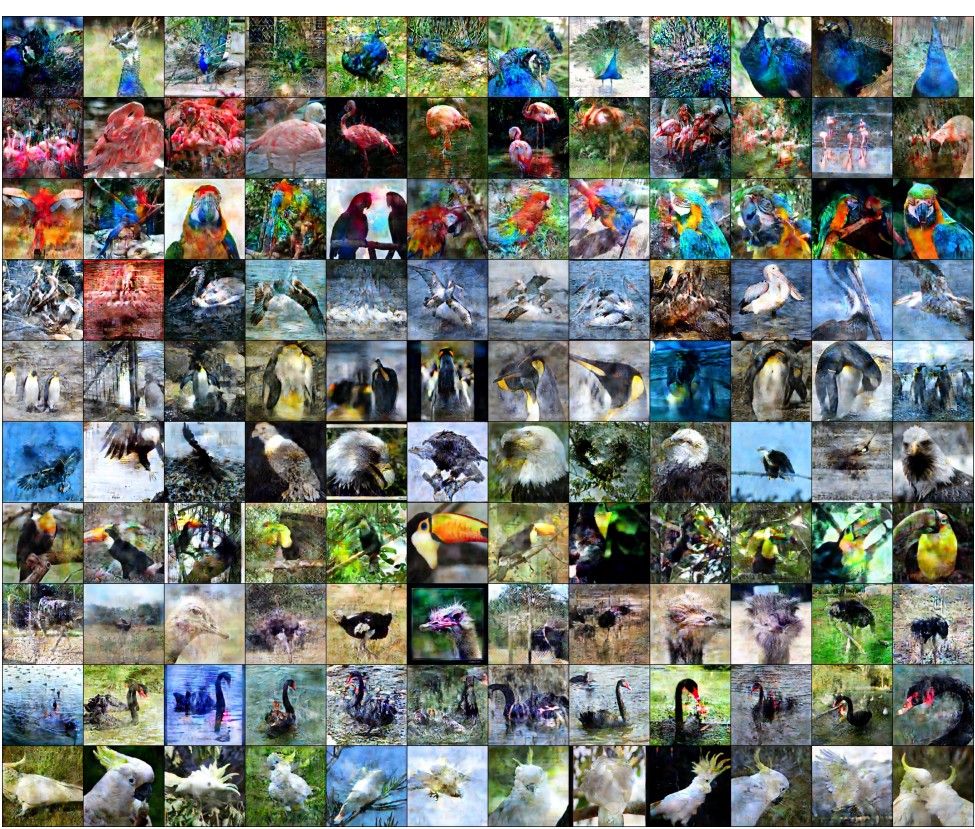

Figure 6: The images decoded from the distilled latent codes on *Bird*, resolution 256, IPC 1, by LatentMTT with $f = 4$.

