# OpenReview forum: "Dataset Distillation in Latent Space"
_ICLR.cc/2024/Conference — ICLR 2024 Conference Withdrawn Submission_

### Official Review · Reviewer_yGia · 2023-11-01

**Soundness:** 3 good
**Presentation:** 2 fair
**Contribution:** 3 good
**Rating:** 6
**Confidence:** 3

**Summary:**

Dataset distillation is known to be expensive in both memory and training time. The authors in this paper propose to address this issue directly through an auto-encoder, where the actual DD only happens in the latent space. The distilled codes can be used to further reconstruct and obtain training images. The authors demonstrate the efficiency and promising training results on the proposed method.

**Strengths:**

+ The proposed method tackles the DD problem from another angle. The latent code distillation makes a lot of sense in terms of efficiency and can potentially help the field on larger datasets
+ The authors demonstrate that the proposed method indeed can achieve descent performance with good efficiency
+ The authors' writing is pretty clear and easy to follow

**Weaknesses:**

- The algorithm seems to be heavily depending on the quality of the pretrained autoencoder, causing another layer of complexity in the distillation procedure.
- In a more general field, language or other modality, where AEs are not that popular, the proposed method can be limited in terms of contribution or usage.
- It seems that the authors only focus on a subset of DD algorithm, how would the latent DD perform using FrePo [1] or momentum-based BPTT [2]? It would be nice if authors can add the comparison and discussion on these two as well.

[1] Dataset Distillation using Neural Feature Regression

[2] Remember the Past: Distilling Datasets into Addressable Memories for Neural Networks

**Questions:**

See above.

---

> ### Author Response · Authors · 2023-11-13
> **Response to Reviewer yGia**
>
> **Q1--Q2**
>
> It is true that the extra autoencoder used by LatentDD may induce additional complexity to dataset distillation (DD) tasks. However, at least for distilling image datasets we can use the off-the-shelf autoencoder provided by stable diffusion. For other modalities (e.g. text, audio, video), though we are not experts, we suppose there should be ways to compress data items into a more info-compact form and then reconstruct them (with acceptable loss or noise) just like autoencoders do to images. In this way, our idea of LatentDD is still applicable to other modalities.
>
> **Q3**
>
> As we mainly transfer three mainstream DD algorithms DC [1], DM [2] and MTT [3] into latent versions, our baseline list also consists of methods following these three base algorithms, which has already included many recent and competitive works such as GLaD [4] and FTD [5].
>
> For the meta-learning methods such as FRePo [6], according to the experimental results reported in Table 3 of [6], the performance gain achieved by FRePo over MTT on ImageNet subsets (resolution 128) is rather marginal, especially when IPC = 1. Taking the results in Table 1 of our paper into regards, we humbly suppose that FRePo is unlikely to outperform LatentMTT. Nevertheless, we agree that combining the idea of LatentDD with meta-learning methods such as FRePo is another interesting topic and we will consider exploring it in future work.
>
>
> **Reference**
>
> [1] Bo Zhao and Hakan Bilen. Dataset condensation with gradient matching. In ICLR, 2021a.
>
> [2] Bo Zhao and Hakan Bilen. Dataset condensation with distribution matching. In WACV, 2023.
>
> [3] George Cazenavette, Tongzhou Wang, Antonio Torralba, Alexei A. Efros, and Jun-Yan Zhu. Dataset distillation by matching training trajectories. In CVPR, 2022.
>
> [4] George Cazenavette, Tongzhou Wang, Antonio Torralba, Alexei A. Efros, and Jun-Yan Zhu. Generalizing dataset distillation via deep generative prior. In CVPR, 2023.
>
> [5] Jiawei Du, Yidi Jiang, Vincent T. F. Tan, Joey Tianyi Zhou, and Haizhou Li. Minimizing the accumulated trajectory error to improve dataset distillation. In CVPR, 2023.
>
> [6] Yongchao Zhou, Ehsan Nezhadarya, and Jimmy Ba. Dataset distillation using neural feature regression. In NeurIPS, 2022.
>
> If you have further questions, please feel free to leave comments.
>
> Authors

---

### Official Review · Reviewer_cyHK · 2023-11-01

**Soundness:** 2 fair
**Presentation:** 2 fair
**Contribution:** 3 good
**Rating:** 3
**Confidence:** 5

**Summary:**

This work aims to address three challenges in Dataset Distillation: high time and space complexities, and low data compactness. They proposed LatentDD to move the distillation from pixel space to latent space, leveraging a pretrained autoencoder from stable diffusion. The LatentDD method significantly reduce time and space requirements in DD tasks, allowing the distillation of higher resolution datasets and offer more info-compact latent codes within the same storage limits.

**Strengths:**

This paper is well-motivated, and well-organized. The observation that "distilling dataset in the original space (e.g. pixel space for image datasets) will inevitably condense high-frequency detailed information into limited storage budget, which is usually unnecessary for downstream tasks" is a solid point to serve as motivation for method design. The authors also provide comprehensive experiments on various datasets.

**Weaknesses:**

1. While this method has demonstrated its effectiveness for high-resolution dataset distillation, there are no experiments and results comparison on lower resolution datasets such as CIFAR10/100. It leaves concern of whether using an autoencoder from stable diffusion for DD impacts the performance of distillation for such datasets.
2. To my knowledge, coreset selection does not belong to dataset distillation, and dataset distillation usually refers to the optimization based methods that distill the data into compact synthetic sets. Therefore, the statement in the introduction: "Some DD methods select a subset from the full dataset according to certain rules (Feldman et al., 2013; Welling, 2009; Sener & Savarese, 2018; Aljundi et al., 2019; Zhou et al., 2023), usually referred to as coreset selection" seems confusing.
3. In the introduction, P1: "computationally intensive bilevel optimization problem" and P2: "space complexity, i.e. DD needs to store the whole computation graph" however, I don't see the present method design that directly aims to address the computation issue of bilevel optimization and I did not see the latentDD method evaluated on bilevel optimization based DD methods. Besides, current method still needs to store the entire computation graph.
4. All three methods chosen by the authors all falls into surrogate objective DD frameworks which seems limiting and the meta-learning based methods are ignored. It would be a lot more convincing to include methods related to meta-learning based DD as well.
5. Eqn.2 is for meta-learning DD framework, and the authors did not evaluate their latentDD method on meta-learning DD framework, which seems kinda disconnected to list Eqn.2 here.
6. The authors claim: "all the previous works have distilled datasets in pixel space" which does not seem accurate. Check [1] for more details.
7. The authors did not report the full dataset (of latent codes) performance, making it hard to evaluate the performance gap between original dataset and distilled ones.
8. Following 7, since the full dataset (of latent codes) performance is unknown, and there's also no experiment with the performance evaluation of the initial latent code (post-autoencoding and pre-distillation), it's unclear to me whether the performance would be good enough just with the latent code itself even without the distillation process. I think it will be interesting to see how much of the performance gain it has during the distillation process, or this distillation process actually hurts the initial latent code's info-compact ability.
9. The cross-architecture results reported in table 6 seems confusing, eg., the performance of ResNet18 (56.00) and VGG11 (49.32) performance are consistently better than the ConvNet results (46.72), even the distilled data comes from ConvNet.
10. The authors only report IPC=1 for Res=512, would be great to see the performance of IPC=10.

[1] Cazenavette, George, et al. "Generalizing Dataset Distillation via Deep Generative Prior." Proceedings of the IEEE/CVF Conference on Computer Vision and Pattern Recognition. 2023.

**Questions:**

Check weaknesses section for more details.

---

> ### Author Response · Authors · 2023-11-13
> **Response to Reviewer cyHK (Part 1)**
>
> **Q1**
>
> We did not include low-resolution datasets in our experiments because our work is focusing on reducing time \& space consumption and improving info-compactness, so that dataset distillation (DD) can be extended to higher resolution or higher image per class (IPC). Therefore we suppose it is less necessary to evaluate on low-resolution datasets wuch as MNIST/SVHN/CIFAR10/CIFAR100, which many previous works can manage already.
>
> As illustrated in Figure 4 of the Appendix, more information loss is observed when applying the autoencoder of stable diffusion to images with lower resolution. It is true that *this* autoencoder may not suit low-resolution datasets, however if there is a pretrained autoencoder available for low-resolution images, our idea of LatentDD is still applicable.
>
> **Q2**
>
> We thank the reviewer for mentioning this point. Previously we did not intend to completely exclude coreset selection methods from DD methods since we suppose selecting a subset is also a kind of distillation. We will further clarify the relation between coreset selection and DD in the revised version.
>
> **Q3**
>
> First we would like to clarify that Problem 1 \& 2 we summarized in Section 1 have already been partially (not entirely) solved by the three mainstream DD algorithms DC [2], DM [3] and MTT [4] by proposing surrogate objectives (e.g. simplified bilevel optimization, and single-step/partial trajectory of the computation graph). Although our LatentDD algorithms did not directly solve Problem 1 \& 2 by further simplifying the bilevel optimization or reducing the computation graph needs to be stored, the goal of LatentDD is to replace pixel-level images with smaller latent codes, so that the computation load and the space needed to store the computation graph can be reduced. In conclusion, Problem 1 \& 2 are first partially solved by DC, DM and MTT, and then further optimized with our idea of LatentDD based on the three aforementioned DD algorithms.
>
> **Q4--5**
>
> To the best of our knowledge, FRePo [5] is one of the state-of-the-art meta-learning DD methods. However, according to the experimental results reported in Table 3 of [5], the performance gain achieved by FRePo over MTT on ImageNet subsets (resolution 128) is rather marginal, especially when IPC = 1. Taking the results in Table 1 of our paper into regards, we humbly suppose that FRePo is unlikely to outperform LatentMTT. Also, as we propose LatentDD algorithms based on DC, DM and MTT, our baseline list mainly consists of succeeding methods based on these three algorithms. Nevertheless, we agree that combining the idea of LatentDD with meta-learning methods such as FRePo is indeed interesting and we will consider exploring this topic in future work.
>
> By the way, Section 3.1 describes the fundamental definition of DD, which is regarded to be useful especially to readers less familiar with DD. Thus we include Equation 2 as the basic and original form of DD. In the following Section 3.2 we have clarified that several algorithms have been proposed to efficiently solving Equation 2, and these algorithms serve as the base methods of LatentDD.
>
> **Q6**
>
> As explained in Section 2 (*Factorization* paragraph), all the previous factorization-based DD methods (including GLaD [6] the reviewer mentioned), are still operating in pixel space when applying DD algorithms. For example, though GLaD tries to distill latent codes within a generative model, it still repeatedly decodes latent codes into pixel-level images before feeding the images into DC, DM or MTT, and backpropagates the gradient from DD algorithms to the latent codes through the generative model (and that's why GLaD DC, GLaD DM and GLaD MTT run much slower than DC, DM and MTT, see Table 2). On the contrary, LatentDD methods directly operate in latent space. The difference between LatentDD and other methods has also been visually illustrated in Figure 1 of our paper.

---

> ### Author Response · Authors · 2023-11-13
> **Response to Reviewer cyHK (Part 2)**
>
> **Q7**
>
> We present the classification accuracy on full set, in pixel space and latent spaces (f = 4 or 8) respectively, in the table below. We will include these results in the revised version.
>
> | Full Set                    | Bird 256 | Fruit 256 | Woof 256 | Cat 256 | Nette 256 | Bird 512 | Fruit 512 |
> | --------------------------- | -------- | --------- | -------- | ------- | --------- | -------- | --------- |
> | Pixel (ConvNetD6)           | 96.40    | 69.40     | 80.00    | 73.60   | 90.40     | 92.20    | 68.80     |
> | Latent ($f = 4$, ConvNetD4) | 92.60    | 71.60     | 77.20    | 73.00   | 91.60     | 95.20    | 75.60     |
> | Latent ($f = 8$, ConvNetD3) | 89.00    | 68.80     | 74.40    | 67.20   | 89.40     | 92.80    | 72.40     |
>
> **Q8**
>
> We present the classification performance with latent codes merely initialized from randomly selected images in the tables below (the first rows of the two tables, following the settings of Table 1 of our paper). These results show that distilling the latent codes are indeed beneficial, though in IPC = 10 the performance gains become less significant probably because there is not much to further improve when we have 120 latent codes per class.
>
> | IPC = 1            | Bird 256 | Fruit 256 | Woof 256 | Cat 256 | Nette 256 | Bird 512 | Fruit 512 |
> | ------------------ | -------- | --------- | -------- | ------- | --------- | -------- | --------- |
> | Rand. Init. Latent | 32.60    | 24.00     | 19.20    | 30.00   | 43.60     | 30.60    | 21.80     |
> | LatentDC           | 46.72    | 30.12     | 28.96    | 38.08   | 55.92     | 47.52    | 29.68     |
> | LatentDM           | 47.08    | 30.68     | 28.00    | 36.28   | 56.08     | 46.20    | 30.60     |
> | LatentMTT          | 52.86    | 37.82     | 39.84    | 41.42   | 62.86     | 52.44    | 36.20     |
>
> | IPC = 10           | Bird 256 | Fruit 256 |
> | ------------------ | -------- | --------- |
> | Rand. Init. Latent | 74.60    | 44.80     |
> | LatentDC           | 80.44    | 51.60     |
> | LatentDM           | 77.20    | 47.76     |
> | LatentMTT          | 78.44    | 52.46     |
>
> **Q9**
>
> As explained in Appendix B.2, since the ConvNet used for distillation (in latent space) and that used for evaluation (in pixel space) have different depth. The results on ConvNet in Table 6 is also cross-architecture to some extent, since in previous works both ConvNets used for distillation and evaluation have identical depths. For this reason, models with more complex architectures (e.g. ResNet18, VGG11) achieve better results than ConvNet.
>
> **Q10**
>
> We did not present the results on resolution 512, IPC = 10 simply because the experiments on every method (including LatentDD and the baselines) take too much time to run. We humbly suppose that the experiments we have currently conducted are already a big leap ahead of previous works to validate the ability of LatentDD methods on high-resolution datasets or larger IPC.
>
> **Reference**
>
> [1] Tongzhou Wang, Jun-Yan Zhu, Antonio Torralba, and Alexei A. Efros. Dataset distillation. arXiv preprint arXiv:1811.10959, 2018.
>
> [2] Bo Zhao and Hakan Bilen. Dataset condensation with gradient matching. In ICLR, 2021a.
>
> [3] Bo Zhao and Hakan Bilen. Dataset condensation with distribution matching. In WACV, 2023.
>
> [4] George Cazenavette, Tongzhou Wang, Antonio Torralba, Alexei A. Efros, and Jun-Yan Zhu. Dataset distillation by matching training trajectories. In CVPR, 2022.
>
> [5] Yongchao Zhou, Ehsan Nezhadarya, and Jimmy Ba. Dataset distillation using neural feature regression. In NeurIPS, 2022.
>
> [6] George Cazenavette, Tongzhou Wang, Antonio Torralba, Alexei A. Efros, and Jun-Yan Zhu. Generalizing dataset distillation via deep generative prior. In CVPR, 2023.
>
> If you have further questions, please feel free to leave comments.
>
> Authors

---

### Official Review · Reviewer_Ya4Y · 2023-11-01

**Soundness:** 2 fair
**Presentation:** 2 fair
**Contribution:** 2 fair
**Rating:** 5
**Confidence:** 4

**Summary:**

This paper addresses the challenges in Dataset Distillation (DD) by transitioning from pixel space to info-compact latent space. This shift reduces time and space requirements while maintaining performance, enabling high-resolution dataset distillation. The paper's method delivers more info-compact latent codes within the same storage constraints, enhancing efficiency.

**Strengths:**

The paper identifies three primary challenges in dataset distillation: high time complexity, high space complexity, and the retention of unnecessary high-frequency information. The authors claim to introduce a pioneering framework that directly addresses these issues by conducting dataset distillation in the latent space, rather than the pixel space.

**Weaknesses:**

● The authors assert that they are the first to successfully address these three challenges in dataset distillation. What specific limitations or hindrances have prevented existing works from generalizing solutions to these problems? Is the proposed method the sole solution, or are there alternative approaches that merit consideration?

● I've noticed that the paper exclusively presents performance experiments on the Sub-ImageNet dataset. Given the existence of prior works that have addressed the full ImageNet condensation problem efficiently, and the authors' claim of efficiency, it would be valuable to see if the authors can tackle this challenging task and report the results.

● How could the third challenge be solved in this framework (distilling dataset in the
original space (e.g. pixel space for image datasets) will inevitably condense high-frequency detailed
information into limited storage budget, which is usually unnecessary for downstream tasks.)? Is it tested on the experiments?

● As far as my current knowledge goes, there are existing works on matching the latent space in the dataset distillation (DD) framework. Could you highlight the primary distinctions between your work and these existing approaches that set your method apart?

**Questions:**

Methodology and Experimental Setup:

a. Could you provide a more detailed description of the methodology used in your experiments, including specific hyperparameters, model architectures, and training protocols?

b. How were the datasets prepared and preprocessed before conducting experiments, and what criteria were used for data selection and cleaning?

Comparative Analysis:

a. In the context of your efficiency claims, can you offer a direct quantitative comparison of your method with existing dataset distillation (DD) frameworks, highlighting the advantages and limitations?

b. Given the broader landscape of DD research, can you elaborate on how your approach compares with other methods in terms of scalability and generalization to different datasets and architectures?

Scalability and Generalization:

a. To address scalability, how does your method perform when applied to datasets with higher resolutions or complex network architectures?

b. Can you discuss the generalization capabilities of your proposed method, particularly in the context of training on models directly (without transfer learning) and its applicability to large-scale datasets like ImageNet?

Latent Space vs. Pixel Space:

a. What are the main advantages of conducting dataset distillation in the latent space, as opposed to the pixel space, and how does this impact the retention of high-frequency information?

b. Could you explain the rationale for not performing experiments on the full ImageNet dataset, given your assertion of efficiency, and how your method could address this challenging task?

Distinguishing Features:

a. In light of existing works that match the latent space in the DD framework, what key differentiating features or innovations characterize your approach?

b. Can you clarify the specific mechanisms or techniques that set your method apart from prior works, leading to the successful resolution of the identified challenges in DD?

---

> ### Author Response · Authors · 2023-11-13
> **Response to Reviewer Ya4Y (Part 1)**
>
> **Methodology and Experimental Setup**
>
> We have already detailed the experimental settings that the reviewer mentions in the Appendix. Specifically, hyperparameters are listed in Appendix A.1, while model architectures and training protocols are in Appendix A.2. We did not elaborate on the detailed designs of ConvNet except for the choice of depth, as it has been used by almost every previous dataset distillation (DD) works and we supposed it would be dull to restate them again in our work. By the way we have also cited the work that proposed ConvNet in the appendix for reference.
>
> With regard to the selection of ImageNet subsets, we have followed previous works MTT [1], GLaD [2] and have specified the source of the subsets in Section 4.1 of the main paper. The preprocessing method is basically a normalize-resize-centercrop procedure, which also follows previous works [1--2] conducting experiments on ImageNet subsets. We will further clarify these parts in the appendix in the revised version.
>
> **Comparative Analysis on Efficiency**
>
> We believe that a comprehensive quantitative comparison on efficiency (time \& space) among LatentDD methods and the baselines have already been shown in Table 2 of the main paper, with analysis in Section 4.2.
>
> **Scalability and Generalization**
>
> Theoretically, our LatentDD methods can be applied to datasets with arbitrarily high resolution, and can adopt any network architecture to replace ConvNet. The only restrictions might be the time consumption and the limitation of GPU memory. Although it is difficult to tell the exact upper bound of the dataset resolution or the network size that LatentDD is capable of (with currently available computing devices), such restrictions has already been largely alleviated since our LatentDD methods have transferred DD processes from pixel space to latent space. Practically, our experiments are conducted on resolution at least 256, which is even higher than the upper bounds of almost all the previous DD works (usually 64, 128 or 224).
>
> Following the reviewer's advice w.r.t. generalization, we present the classification accuracy on full set, in pixel space and latent spaces (f = 4 or 8) respectively, in the table below. We may observe that
> - Generally speaking, the performance in latent space is comparable to pixel space without significant gap.
> - When the resolution is lower (256), the performance in latent space may be slightly inferior than pixel space. A possible reason is that encoding images into latent codes has induced some information loss.
> - However when the resolution is high (512), latent space outperforms pixel space, probably due to less information loss.
> - Comparing resolution 256 with 512, the performance in pixel space slightly drops, while that in latent space improves a little. It may because that high-resolution pixel-level images have too much redundant and high-frequency information that is less useful to classifiaction tasks.
> - Comparing f = 4 with 8, the performance of f = 4 is better probably because larger downsampling factor will induce more information loss.
>
> From the analysis above, we may conclude that the generalization of transferring classification tasks (and also DD for classification tasks) from pixel space to latent space can be assured (at least on high-resolution cases where encoding images into latent codes will not lose too much information).
>
> | Full Set                    | Bird 256 | Fruit 256 | Woof 256 | Cat 256 | Nette 256 | Bird 512 | Fruit 512 |
> | --------------------------- | -------- | --------- | -------- | ------- | --------- | -------- | --------- |
> | Pixel (ConvNetD6)           | 96.40    | 69.40     | 80.00    | 73.60   | 90.40     | 92.20    | 68.80     |
> | Latent ($f = 4$, ConvNetD4) | 92.60    | 71.60     | 77.20    | 73.00   | 91.60     | 95.20    | 75.60     |
> | Latent ($f = 8$, ConvNetD3) | 89.00    | 68.80     | 74.40    | 67.20   | 89.40     | 92.80    | 72.40     |
>
>
> About running on the full ImageNet, we have noticed that there are indeed some previous works (yet only a few) conducting experiments under this setting, on low resolution of 64. We suppose none of the previous works can manage resolution 256 or 512 on the full ImageNet, since prebuilding the datasets itself will take hundreds of gigabytes of memory. However, our LatentDD might be capable of such setting as we only store the latent codes encoded from real dataset images and thus the memory consumption is 12 (f = 4) or 48 (f = 8) times smaller. Besides, as we have already evaluated our LatentDD algorithms on several ImageNet subsets that are commonly used by previous works (e.g. MTT [1], GLaD [2]), we suppose these experimental results are enough to verify the ability of LatentDD.

---

> ### Author Response · Authors · 2023-11-13
> **Response to Reviewer Ya4Y (Part 2)**
>
> **Latent Space vs. Pixel Space**
>
> As detailedly described in Section 1 and Section 3.2, and empirically verified by experiments in Section 4, the main advantages of conducting DD in latent space instead of pixel space are that LatentDD methods can settle the three main problems in current DD researches (time consumption, space consumption and info-compactness), mentioned in Section 1.
>
> The retention of high-frequency information in encoding/decoding images and latent codes has been discussed in the paper of latent diffusion models [3] where the autoencoder we use for LatentDD was proposed, we suggest referring to [3] for details. Figure 4 in the Appendix of our work has also proved the retention of high-frequency by visual examples. The reconstructed images keep the low-frequency information (e.g. structures) of the original images while lose some high-frequency information (e.g. details). This phenomenon is especially obvious when resolution is low. By the way, the main results in our paper and the performance on full set in the table above have also proved that such retention of high-frequency information does not harm classification tasks (and also DD for classification tasks).
>
> **Distinguishing Features**
>
> We suppose the reviewer refers to the feature matching methods (see Section 2) DM [4], IDM [5] and CAFE [6] by mentioning *existing works that match the latent space*. While these feature matching methods are seeking surrogate objectives of the original meta-learning DD method [7], our LatentDD should be categorized as a factorization method (see Section 2). The key difference is, instead of proposing DD algorithms optimizing surrogate objectives while still operating on pixel space as [4--6], LatentDD actually transfers mainstream DD algorithms from pixel space to latent space, in order to solve the three problems mentioned in the paper (time consumption, space consumption and info-compactness) that are commonly observed in previous DD methods (including the three feature matching methods [4--6]). Therefore, the existing feature matching methods and our LatentDD are following two totally different research directions in the area of DD. Also, LatentDM, as one of the LatentDD methods, is a combination of DM [4] and our idea of transferring DD to latent space.
>
> With regard to the specific mechanisms and techniques of our work that settle the three challenges in DD, we have already described in Section 1, Section 2 (*Factorization* paragraph) and Section 4. We further succinctly summarize the key designs solving the three problems when compared with previous works below, for a quick lookup.
>
> - P1 \& P2 (time and space consumption): Running DD algorithms in laten space instead of pixel space can accelerate the computation and reduce the memory needed to store the real and the synthetic samples, with only marginal cost of performance. Compared with previous factorization methods which still operate on pixel-level images, LatentDD algorithms can get rid of repeatedly decoding latent codes into images and backpropagating gradient from images to latent codes during the DD processes.
> - P3 (info-compactness): LatentDD algorithms deliver highly info-compact latent codes to downstream tasks instead of the less info-compact pixel-level images. In this way LatentDD can deliver more information within in the same storage budget.
>
> **Reference**
>
> [1] George Cazenavette, Tongzhou Wang, Antonio Torralba, Alexei A. Efros, and Jun-Yan Zhu. Dataset distillation by matching training trajectories. In CVPR, 2022.
>
> [2] George Cazenavette, Tongzhou Wang, Antonio Torralba, Alexei A. Efros, and Jun-Yan Zhu. Generalizing dataset distillation via deep generative prior. In CVPR, 2023.
>
> [3] Robin Rombach, Andreas Blattmann, Dominik Lorenz, Patrick Esser, and Bjorn Ommer. Highresolution image synthesis with latent diffusion models. In CVPR, 2022.
>
> [4] Bo Zhao and Hakan Bilen. Dataset condensation with distribution matching. In WACV, 2023.
>
> [5] Ganlong Zhao, Guanbin Li, Yipeng Qin, and Yizhou Yu. Improved distribution matching for dataset condensation. In CVPR, 2023.
>
> [6] Kai Wang, Bo Zhao, Xiangyu Peng, Zheng Zhu, Shuo Yang, Shuo Wang, Guan Huang, Hakan Bilen, Xinchao Wang, and Yang You. CAFE: Learning to condense dataset by aligning features. In CVPR, 2022.
>
> [7] Tongzhou Wang, Jun-Yan Zhu, Antonio Torralba, and Alexei A. Efros. Dataset distillation. arXiv preprint arXiv:1811.10959, 2018.
>
> If you have further questions, please feel free to leave comments.
>
> Authors

---

### Official Review · Reviewer_KLWs · 2023-11-06

**Soundness:** 3 good
**Presentation:** 3 good
**Contribution:** 2 fair
**Rating:** 3
**Confidence:** 5

**Summary:**

This paper proposes to perform dataset distillation in the latent space instead of the pixel space. The proposed method first encodes real images of target dataset into the latent codes. Then three representative (pixel level) distillation methods are adapted to distill latent codes. After that, the distilled latent codes are fed into the decoder to get the distilled images.

**Strengths:**

This paper shows that, performing the distillation in latent space costs less resources than the distillation in pixel space, without sacrificing much performance.

**Weaknesses:**

Most comparisons in this paper are UNFAIR.

In dataset distillation area, previous works compare the performance under the same IPC (image per class) settings, which means that the AMOUNT of the distilled images fed into the evaluation network is fixed.

This paper proposes ‘LPC’ (latent per class) and claims that 1 IPC=12 LPC since their size are the same (latent codes have lower resolution). Then the authors compare their method’s performance (12*n LPC) with previous works (n IPC), which means that the method proposed in this paper actually uses twelve times more images than previous methods for evaluation.

I think comparing performances under the same ‘storage consumption’ settings rather than IPC are unfair and unacceptable. Otherwise, we can perform the distillation first and then use an auto-encoder to compress the distilled images, such that we can use more distilled data under the same ‘storage consumption’ setting; accordingly, the performance is improved. Then the development of dataset distillation might turns toward finding a stronger auto-encoder.

**Questions:**

In Table 2, the time consumption of LatentDC/DM/MTT is evaluated under the same IPC settings or the proposed LPC settings?

I think the selling point of this paper should be: Performing distillation in latent space is quicker, low-cost, and will not sacrificing performance drastically. It is fine to perform worse than previous works since it is hard to acquire lower cost and better performance at the same time. Please stop performing the evaluation under ‘LPC’ settings. I suggest the authors to perform a fair comparison and highlight their contributions better (such as low cost).

---

> ### Author Response · Authors · 2023-11-12
> **Response to Reviewer KLWs**
>
> **Fairness Issue**
>
> We thank the reviewer for mentioning the fairness issue in dataset distillation (DD). However we are afraid that we cannot agree with the point that performing DD under the same storage budget is unfair or unacceptable, since how to efficiently encode more information in a limited storage consumption has always been an important research topic (see *Factorization* paragraph of Section 2), which is orthogonal to other topics such as surrogate objectives and optimization. Actually there are a plenty of previous works trying to break the restraint of delivering less info-compact pixel-level images in DD. We list some of the examples below.
>
> IDC [1] divides one full-size image of resolution $H \times W$ into $f \times f$ sub-images with resolution $(H / f) \times (W / f)$ with a downsampling factor $f$. As $f$ is usually 2, 3 or 4, IDC encodes 4, 9 or 16 sub-images within the storage budget of one full-size image. These sub-images are resized to full size before being fed into DD algorithms, but still stored in their compressed size when delivered. Therefore IDC is actually running on 4, 9 or 16 sub-images per class (similar to our LPC) when IPC = 1.
>
> HaBa [2] factorizes pixel-level images into $n$ base images (similar to latent codes) and $m$ hallucinators (similar to decoders), so that a total of $n \times m$ full-size images can be delivered by feeding these base images to the hallucinators. Practically, within the same storage budget of IPC = 10 in a 10-class classification, HaBa saves $n = 9$ base images per class and $m = 5$ hallucinators shared among classes, resulting in 45 images per class in fact. Similar idea of latent codes + decoders is also explored in KFS [3], where LPC = 16 $\times$ IPC in their settings.
>
> RTP [4] transforms the retrieval of distilled images into a query of addressing matrices and common bases shared among classes. Under one of the possible settings, RTP stores 20 bases and 76 addressing matrices within the same storage of IPC = 1, which means there are actually 76 images per class when evaluated on classification tasks.
>
> Besides the previous works above that seek efficient ways to store information within the same storage budgets in DD. There are also many recent works which are based on the sub-image idea of IDC (e.g. IDM [5], DREAM [6], YOCO [7]).
>
> Among these works, we have already mentioned and cited [1--6] in our paper ([7] is published after this submission) and also have included IDC [1] into our baseline list, as a popular DD method. In conclusion, we suppose that the research topic of efficiently encoding more information into limited storage cannot be ignored, since the info-compactness mentioned in our work is probably the most critical factor that affect the performance of DD.
>
> **About Table 2**
>
> The time consumption is evaluated under the LPC setting, which means 1 IPC = 12 LPC and 10 IPC = 120 LPC, same as the main results in Table 1. We will clarify this setting in revised version.
>
> **Reference**
>
> [1] Jang-Hyun Kim, Jinuk Kim, Seong Joon Oh, Sangdoo Yun, Hwanjun Song, Joonhyun Jeong, Jung-Woo Ha, and Hyun Oh Song. Dataset condensation via efficient synthetic-data parameterization. In ICML, 2022.
>
> [2] Songhua Liu, Kai Wang, Xingyi Yang, Jingwen Ye, and Xinchao Wang. Dataset distillation via factorization. In NeurIPS, 2022.
>
> [3] Hae Beom Lee, Dong Bok Lee, and Sung Ju Hwang. Dataset condensation with latent space knowledge factorization and sharing. arXiv preprint arXiv:2208.00719, 2022a.
>
> [4] Zhiwei Deng and Olga Russakovsky. Remember the past: Distilling datasets into addressable memories for neural networks. In NeurIPS, 2022.
>
> [5] Ganlong Zhao, Guanbin Li, Yipeng Qin, and Yizhou Yu. Improved distribution matching for dataset condensation. In CVPR, 2023.
>
> [6] Yanqing Liu, Jianyang Gu, Kai Wang, Zheng Zhu, Wei Jiang, and Yang You. DREAM: Efficient dataset distillation by representative matching. In ICCV, 2023.
>
> [7] Yang He, Lingao Xiao, and Joey Tianyi Zhou. You only condense once: Two rules for pruning condensed datasets. In NeurIPS, 2023.
>
>
> If you have further questions, please feel free to leave comments.
>
> Authors

---

### Author Response · Authors · 2023-11-12
**To all the reviewers**

Dear reviewers,

First we would like to express our sincere gratitude for the insightful comments raised by the reviewers. Later on we will reply to reviewers' concerns separately.

Authors